# Diverse Sequential Subset Selection for Supervised Video Summarization

**Boqing Gong**[*]
Department of Computer Science
University of Southern California
Los Angeles, CA 90089
`boqinggo@usc.edu`

**Wei-Lun Chao**[*]
Department of Computer Science
University of Southern California
Los Angeles, CA 90089
`weilunc@usc.edu`

**Kristen Grauman**
Department of Computer Science
University of Texas at Austin
Austin, TX 78701
`grauman@cs.utexas.edu`

**Fei Sha**
Department of Computer Science
University of Southern California
Los Angeles, CA 90089
`feisha@usc.edu`

## Abstract

Video summarization is a challenging problem with great application potential. Whereas prior approaches, largely unsupervised in nature, focus on sampling useful frames and assembling them as summaries, we consider video summarization as a supervised subset selection problem. Our idea is to teach the system to learn from human-created summaries how to select informative and diverse subsets, so as to best meet evaluation metrics derived from human-perceived quality. To this end, we propose the *sequential determinantal point process* (seqDPP), a probabilistic model for diverse sequential subset selection. Our novel seqDPP heeds the inherent sequential structures in video data, thus overcoming the deficiency of the standard DPP, which treats video frames as randomly permutable items. Meanwhile, seqDPP retains the power of modeling diverse subsets, essential for summarization. Our extensive results of summarizing videos from 3 datasets demonstrate the superior performance of our method, compared to not only existing unsupervised methods but also naive applications of the standard DPP model.

## 1 Introduction

It is an impressive yet alarming fact that there is far more video being captured—by consumers, scientists, defense analysts, and others—than can ever be watched or browsed efficiently. For example, 144,000 hours of video are uploaded to YouTube daily; lifeloggers with wearable cameras amass Gigabytes of video daily; 422,000 CCTV cameras perched around London survey happenings in the city 24/7. With this explosion of video data comes an ever-pressing need to develop *automatic video summarization* algorithms. By taking a long video as input and producing a short video (or keyframe sequence) as output, video summarization has great potential to reign in raw video and make it substantially more browseable and searchable.

Video summarization methods often pose the problem in terms of *subset selection*: among all the frames (subshots) in the video, which key frames (subshots) should be kept in the output summary? There is a rich literature in computer vision and multimedia developing a variety of ways to answer this question [1, 2, 3, 4, 5, 6, 7, 8, 9, 10]. Existing techniques explore a plethora of properties that a good summary should capture, designing criteria that the algorithm should prioritize when deciding

---

[*]Equal contribution

which subset of frames (or subshots) to select. These include *representativeness* (the frames should depict the main contents of the videos) [1, 2, 10], *diversity* (they should not be redundant) [4, 11], *interestingness* (they should have salient motion/appearance [2, 3, 6] or trackable objects [5, 12, 7]), or *importance* (they should contain important objects that drive the visual narrative) [8, 9].

Despite valuable progress in developing the desirable properties of a summary, prior approaches are impeded by their unsupervised nature. Typically the selection algorithm favors extracting content that satisfies criteria like the above (diversity, importance, etc.), and performs some sort of frame clustering to discover events. Unfortunately, this often requires some hand-crafting to combine the criteria effectively. After all, the success of a summary ultimately depends on human perception. Furthermore, due to the large number of possible subsets that could be selected, it is difficult to directly optimize the criteria *jointly* on the selected frames as a subset; instead, sampling methods that identify independently useful frames (or subshots) are common.

To address these limitations, we propose to consider video summarization as a *supervised* subset selection problem. The main idea is to use examples of human-created summaries—together with their original source videos—to teach the system how to select informative subsets. In doing so, we can escape the hand-crafting often necessary for summarization, and instead directly optimize the (learned) factors that best meet evaluation metrics derived from human-perceived quality. Furthermore, rather than independently select "high scoring" frames, we aim to capture the interlocked dependencies between a given frame and all others that could be chosen.

To this end, we propose the *sequential determinantal point process* (seqDPP), a new probabilistic model for sequential and diverse subset selection. The determinantal point process (DPP) has recently emerged as a powerful method for selecting a diverse subset from a "ground set" of items [13], with applications including document summarization [14] and information retrieval [15]. However, existing DPP techniques have a fatal modeling flaw if applied to video (or documents) for summarization: they fail to capture their inherent sequential nature. That is, a standard DPP treats the inputs as *bags of randomly permutable items* agnostic to any temporal structure. Our novel seqDPP overcomes this deficiency, making it possible to faithfully represent the temporal dependencies in video data. At the same time, it lets us pose summarization as a supervised learning problem.

While learning how to summarize from examples sounds appealing, why should it be possible— particularly if the input videos are expected to vary substantially in their subject matter?[1] Unlike more familiar supervised visual recognition tasks, where test data can be reasonably expected to look like the training instances, a supervised approach to video summarization must be able to learn generic properties that transcend the specific content of the training set. For example, the learner can recover a "meta-cue" for representativeness, if the input features record profiles of the similarity between a frame and its increasingly distant neighbor frames. Similarly, category-independent cues about an object's placement in the frame, the camera person's active manipulation of viewpoint/zoom, etc., could play a role. In any such case, we can expect the learning algorithm to focus on those meta-cues that are shared by the human-selected frames in the training set, even though the subject matter of the videos may differ.

In short, our main contributions are: a novel learning model (seqDPP) for selecting diverse subsets from a sequence, its application to video summarization (the model is applicable to other sequential data as well), an extensive empirical study with three benchmark datasets, and a successful first-step/proof-of-concept towards using human-created video summaries for learning to select subsets.

The rest of the paper is organized as follows. In section 2, we review DPP and its application to document summarization. In section 3, we describe our seqDPP method, followed by a discussion of related work in section 4. We report results in section 5, then conclude in section 6.

## 2 Determinantal point process (DPP)

The DPP was first used to characterize the Pauli exclusion principle, which states that two identical particles cannot occupy the same quantum state simultaneously [16]. The notion of exclusion has made DPP an appealing tool for modeling diversity in application such as document summarization [14, 13], or image search and ranking [17]. In what follows, we give a brief account on DPP and how to apply it to document summarization where the goal is to generate a summary by selecting

several sentences from a long document [18, 19]. The selected sentences should be not only diverse (i.e., different) to reduce the redundancy in the summary, but also representative of the document.

**Background** Let $\mathcal{Y} = \{1, 2, \cdots, \mathsf{N}\}$ be a ground set of $\mathsf{N}$ items (eg., sentences). In its simplest form, a DPP defines a discrete probability distribution over all the $2^{\mathsf{N}}$ subsets of $\mathcal{Y}$. Let $Y$ denote the random variable of selecting a subset. $Y$ is then distributed according to

$$P(Y = \boldsymbol{y}) = \frac{\det(\boldsymbol{L_y})}{\det(\boldsymbol{L} + \boldsymbol{I})} \tag{1}$$

for $\boldsymbol{y} \subseteq \mathcal{Y}$. The kernel $\boldsymbol{L} \in \mathbb{S}_+^{\mathsf{N} \times \mathsf{N}}$ is the DPP's parameter and is constrained to be positive semidefinite. $\boldsymbol{I}$ is the identity matrix. $\boldsymbol{L_y}$ is the principal minor (sub-matrix) with rows and columns selected according to the indices in $\boldsymbol{y}$. The determinant function $\det(\cdot)$ gives rise to the interesting property of pairwise repulsion. To see that, consider selecting a subset of two items $i$ and $j$. We have

$$P(Y = \{i, j\}) \propto L_{ii}L_{jj} - L_{ij}^2. \tag{2}$$

If the items $i$ and $j$ are the same, then $P(Y = \{i, j\}) = 0$ (because $L_{ij} = L_{ii} = L_{jj}$). Namely, identical items should not appear together in the same set. A more general case also holds: if $i$ and $j$ are similar to each other, then the probability of observing $i$ and $j$ in a subset together is going to be less than that of observing either one of them alone (see the excellent tutorial [13] for details).

The most diverse subset of $\mathcal{Y}$ is thus the one that attains the highest probability

$$\boldsymbol{y}^* = \arg\max_{\boldsymbol{y}} P(Y = \boldsymbol{y}) = \arg\max_{\boldsymbol{y}} \det(\boldsymbol{L_y}), \tag{3}$$

where $\boldsymbol{y}^*$ results from MAP inference. This is a NP-hard combinatorial optimization problem. However, there are several approaches to obtaining approximate solutions [13, 20].

**Learning DPPs for document summarization** Suppose we model selecting a subset of sentences as a DPP over all sentences in a document. We are given a set of training samples in the form of documents (i.e., ground sets) and the ground-truth summaries. How can we discover the underlying parameter $\boldsymbol{L}$ so as to use it for generating summaries for new documents?

Note that the new documents will likely have sentences that have not been seen before in the training samples. Thus, the kernel matrix $\boldsymbol{L}$ needs to be reparameterized in order to generalize to unseen documents. [14] proposed a special reparameterization called quality/diversity decomposition:

$$\boldsymbol{L}_{ij} = q_i \phi_i^{\mathsf{T}} \phi_j q_j, \quad q_i = \exp\left(\frac{1}{2}\boldsymbol{\theta}^{\mathsf{T}}\boldsymbol{x}_i\right), \tag{4}$$

where $\phi_i$ is the normalized TF-IDF vector of the sentence $i$ so that $\phi_i^{\mathsf{T}}\phi_j$ computes the cosine angle between two sentences. The "quality" feature vector $\boldsymbol{x}_i$ encodes the contextual information about $i$ and its representativeness of other items. In document summarization, $\boldsymbol{x}_i$ are the sentence lengths, positions of the sentences in the texts, and other *meta cues*. The parameter $\boldsymbol{\theta}$ is then optimized with maximum likelihood estimation (MLE) such that the target subsets have the highest probabilities

$$\boldsymbol{\theta}^* = \arg\max_{\boldsymbol{\theta}} \sum_n \log P(Y = \boldsymbol{y}_n^*; \boldsymbol{L}_n(\boldsymbol{\theta})), \tag{5}$$

where $\boldsymbol{L}_n$ is the $\boldsymbol{L}$ matrix formulated using sentences in the $n$-th ground set, and $\boldsymbol{y}_n^*$ is the corresponding ground-truth summary.

Despite its success in document summarization [14], a direct application of DPP to video summarization is problematic. The DPP model is agnostic about the order of the items. For video (and to a large degree, text data), it does not respect the inherent sequential structures. The second limitation is that the quality-diversity decomposition, while cleverly leading to a convex optimization, limits the power of modeling complex dependencies among items. Specifically, only the quality factor $q_i$ is optimized on the training data. We develop new approaches to overcoming those limitations.

## 3 Approach

In what follows, we describe our approach for video summarization. Our approach contains three components: (1) a preparatory yet crucial step that generates ground-truth summaries from multiple human-created ones (section 3.1); (2) a new probabilistic model—the sequential determinantal point process (seqDPP)—that models the process of sequentially selecting diverse subsets (section 3.2); (3) a novel way of re-parameterizing seqDPP that enables learning more flexible and powerful representations for subset selection from standard visual and contextual features (section 3.3).

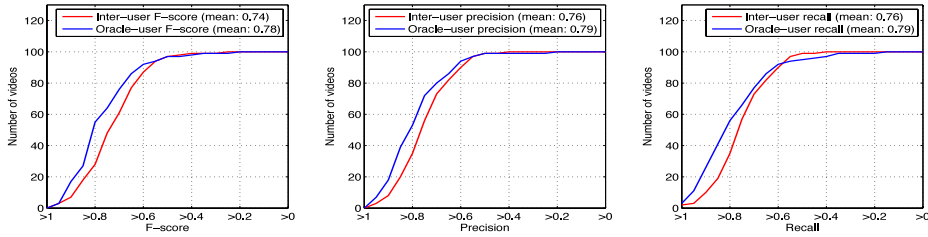

Figure 1: The agreement among human-created summaries is high, as is the agreement between the oracle summary generated by our algorithm (cf. section 3.1) and human annotations.

## 3.1 Generating ground-truth summaries

The first challenge we need to address is what to provide to our learning algorithm as ground-truth summaries. In many video datasets, each video is annotated (manually summarized) by multiple human users. While the users were often well instructed on the annotation task, discrepancies are expected due to many uncontrollable individual factors such as whether the person was attentive, idiosyncratic viewing preferences, etc. There are some studies on how to evaluate automatically generated summaries in the presence of multiple human-created annotations [21, 22, 23]. However, for learning, our goal is to generate one single ground-truth or "oracle" summary per video.

Our main idea is to synthesize the oracle summary that maximally agrees with all annotators. Our hypothesis is that despite the discrepancies, those summaries nonetheless share the common traits of reflecting the subject matters in the video. These commonalities, to be discovered by our synthesis algorithm, will provide *strong enough* signals for our learning algorithm to be successful.

To begin with, we first describe a few metrics in quantifying the agreement in the simplest setting where there are only two summaries. These metrics will also be used later in our empirical studies to evaluate various summarization methods. Using those metrics, we then analyze the consistency of human-created summaries in two video datasets to validate our hypothesis. Finally, we present our algorithm for synthesizing one single oracle summary per video.

**Evaluation metrics** Given two video summaries A and B, we measure how much they are in agreement by first matching their frames, as they might be of different lengths. Following [24], we compute the pairwise distances between all frames across the two summaries. Two frames are then "matched" if their visual difference is below some threshold; a frame is constrained to appear in the matched pairs at most once. After the matching, we compute the following metrics (commonly known as Precision, Recall and F-score):

$$P_{AB} = \frac{\#\text{matched frames}}{\#\text{frames in A}}, \quad R_{AB} = \frac{\#\text{matched frames}}{\#\text{frames in B}}, \quad F_{AB} = \frac{P_{AB} \cdot R_{AB}}{0.5(P_{AB} + R_{AB})}.$$

All of them lie between 0 and 1, and higher values indicate better agreement between A and B. Note that these metrics are not symmetric – if we swap A and B, the results will be different.

Our idea of examining the consistency among all summaries is to treat each summary in turn as if it were the gold-standard (and assign it as B) while treating the other summaries as A's. We report our analysis of existing video datasets next.

**Consistency in existing video databases** We analyze video summaries in two video datasets: 50 videos from the Open Video Project (OVP) [25] and another 50 videos from Youtube [24]. Details about these two video datasets are in section 5. We briefly point out that the two datasets have very different subject matters and composition styles. Each of the 100 videos has 5 annotated summaries. For each video, we compute the pairwise evaluation metrics in precision, recall, and F-score by forming total 20 pairs of summaries from two different annotators. We then average them per video. We plot how these averaged metrics distribute in Fig. 1. The plots show the number of videos (out of 100) whose averaged metrics exceed certain thresholds, marked on the horizontal axes. For example, more than 80% videos have an averaged F-score greater than 0.6, and 60% more than 0.7. Note that there are many videos ($\approx$20) with averaged F-scores greater than 0.8, indicating that on average, human-created summaries have a high degree of agreement. Note that the mean values of the averaged metrics per video are also high.

**Greedy algorithm for synthesizing an oracle summary**    Encouraged by our findings, we develop a greedy algorithm for synthesizing one oracle summary per video, from multiple human-created ones. This algorithm is adapted from a similar one for document summarization [14]. Specifically, for each video, we initialize the oracle summary with the empty set $\boldsymbol{y}^* = \emptyset$. Iteratively, we then add to $\boldsymbol{y}^*$ one frame $i$ at a time from the video sequence

$$\boldsymbol{y}^* \leftarrow \boldsymbol{y}^* \cup \; \arg\max_i \sum_u F_{\boldsymbol{y}^* \cup i, \boldsymbol{y}_u}. \tag{6}$$

In words, the frame $i$ is selected to maximally increase the F-score between the new oracle summary and the human-created summaries $\boldsymbol{y}_u$. To avoid adding all frames in the video sequence, we stop the greedy process as soon as there is no frame that can increase the F-score.

We measure the quality of the synthesized oracle summaries by computing their mean agreement with the human annotations. The results are shown in Fig. 1 too. The quality is high: more than 90% of the oracle summaries agree well with other summaries, with an F-score greater than 0.6. In what follows, we will treat the oracle summaries as ground-truth to inform our learning algorithms.

### 3.2   Sequential determinantal point processes (seqDPP)

The determinantal point process, as described in section 2, is a powerful tool for modeling diverse subset selection. However, video frames are more than items in a set. In particular, in DPP, the ground set is a bag – items are randomly permutable such that the most diverse subset remains unchanged. Translating this into video summarization, this modeling property essentially suggests that we could randomly shuffle video frames and expect to get the same summary!

To address this serious deficiency, we propose sequential DPP, a new probabilistic model to introduce strong dependency structures between items. As a motivating example, consider a video portraying the sequence of someone leaving home for school, coming back to home for lunch, leaving for market and coming back for dinner. If only visual appearance cues are available, a vanilla DPP model will likely select only one frame from the home scene and repel other frames occurring at the home. Our model, on the other hand, will recognize that the temporal span implies those frames are still *diverse* despite their visual similarity. Thus, our modeling intuition is that *diversity should be a weaker prior for temporally distant frames but ought to act more strongly for closely neighboring frames*. We now explain how our seqDPP method implements this intuition.

**Model definition**    Given a ground set (a long video sequence) $\mathcal{Y}$, we partition it into $T$ disjoint yet consecutive short segments $\bigcup_{t=1}^{T} \mathcal{Y}_t = \mathcal{Y}$. At time $t$, we introduce a subset selection variable $Y_t$. We impose a DPP over two neighboring segments where the ground set is $\boldsymbol{U}_t = \mathcal{Y}_t \cup \boldsymbol{y}_{t-1}$, ie., the union between the video segments and the *selected subset* in the immediate past. Let $\boldsymbol{\Omega}_t$ denote the $\boldsymbol{L}$-matrix defined over the ground set $\boldsymbol{U}_t$. The conditional distribution of $Y_t$ is thus given by,

$$P(Y_t = \boldsymbol{y}_t | Y_{t-1} = \boldsymbol{y}_{t-1}) = \frac{\det \boldsymbol{\Omega}_{\boldsymbol{y}_{t-1} \cup \boldsymbol{y}_t}}{\det(\boldsymbol{\Omega}_t + \boldsymbol{I}_t)}. \tag{7}$$

As before, the subscript $\boldsymbol{y}_{t-1} \cup \boldsymbol{y}_t$ selects the corresponding rows and columns from $\boldsymbol{\Omega}_t$. $\boldsymbol{I}_t$ is a diagonal matrix, the same size as $\boldsymbol{U}_t$. However, the elements corresponding to $\boldsymbol{y}_{t-1}$ are zeros and the elements corresponding to $\mathcal{Y}_t$ are 1 (see [13] for details). Readers who are familiar with DPP might identify the conditional distribution is also a DPP, restricted to the ground set $\mathcal{Y}_t$.

The conditional probability is defined in such a way that at time $t$, the subset selected should be diverse among $\mathcal{Y}_t$ as well as be diverse from previously selected $\boldsymbol{y}_{t-1}$. However, beyond those two priors, the subset is not constrained by subsets selected in the distant past. Fig. 2 illustrates the idea in graphical model notation. In particular, the joint distribution of all subsets is factorized

$$P(Y_1 = \boldsymbol{y}_1, Y_2 = \boldsymbol{y}_2, \cdots, Y_T = \boldsymbol{y}_T) = P(Y_1 = \boldsymbol{y}_1) \prod_{t=2} P(Y_t = \boldsymbol{y}_t | Y_{t-1} = \boldsymbol{y}_{t-1}). \tag{8}$$

**Inference and learning**    The MAP inference for the seqDPP model eq. (8) is as hard as the standard DPP model. Thus, we propose to use the following online inference, analogous to Bayesian belief updates (for Kalman filtering):

$$\boldsymbol{y}_1^* = \arg\max_{\boldsymbol{y} \in \mathcal{Y}_1} P(Y_1 = \boldsymbol{y}) \qquad\qquad \boldsymbol{y}_2^* = \arg\max_{\boldsymbol{y} \in \mathcal{Y}_2} P(Y_2 = \boldsymbol{y} | Y_1 = \boldsymbol{y}_1^*) \quad \cdots$$

$$\boldsymbol{y}_t^* = \arg\max_{\boldsymbol{y} \in \mathcal{Y}_t} P(Y_t = \boldsymbol{y} | Y_{t-1} = \boldsymbol{y}_{t-1}^*) \quad \cdots \cdots$$

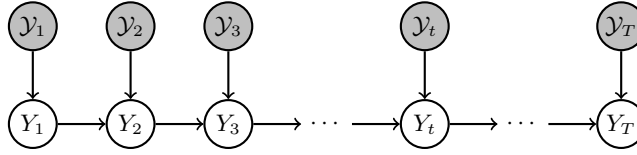

Figure 2: Our sequential DPP for modeling sequential video data, drawn as a Bayesian network

Note that, at each step, the ground set could be quite small; thus an exhaustive search for the most diverse subset is plausible. The parameter learning is similar to the standard DPP model. We describe the details in the supplementary material.

### 3.3 Learning representations for diverse subset selection

As described previously, the kernel $L$ of DPP hinges on the reparameterization with features of the items that can generalize across different ground sets. The quality-diversity decomposition in eq. (4), while elegantly leading to convex optimization, is severely limited in its power in modeling complex items and dependencies among them. In particular, learning the subset selection rests solely on learning the quality factor, as the diversity component remains handcrafted and fixed.

We overcome this deficiency with more flexible and powerful representations. Concretely, let $\boldsymbol{f}_i$ stand for the feature representation for item (frame) $i$, including both low-level visual cues and meta-cues such as contextual information. We reparameterize the $L$ matrix with $\boldsymbol{f}_i$ in two ways.

**Linear embeddings**  The simplest way is to linearly transform the original features

$$L_{ij} = \boldsymbol{f}_i^T \boldsymbol{W}^{\mathrm{T}} \boldsymbol{W} \boldsymbol{f}_j, \tag{9}$$

where $\boldsymbol{W}$ is the transformation matrix.

**Nonlinear hidden representation**  We use a one-hidden-layer neural network to infer a hidden representation for $\boldsymbol{f}_i$

$$L_{ij} = \boldsymbol{z}_i^T \boldsymbol{W}^{\mathrm{T}} \boldsymbol{W} \boldsymbol{z}_j \quad \text{where} \quad \boldsymbol{z}_i = \tanh(\boldsymbol{U} \boldsymbol{f}_i), \tag{10}$$

where $\tanh(\cdot)$ stands for the hyperbolic transfer function.

To learn the parameters $\boldsymbol{W}$ or $\boldsymbol{U}$ and $\boldsymbol{W}$, we use maximum likelihood estimation (cf. eq. (5)), with gradient-descent to optimize. Details are given in the supplementary material.

## 4  Related work

Space does not permit a thorough survey of video summarization methods. Broadly speaking, existing approaches develop a variety of selection criteria to prioritize frames for the output summary, often combined with temporal segmentation. Prior work often aims to retain diverse and representative frames [2, 1, 10, 4, 11], and/or defines novel metrics for object and event saliency [3, 2, 6, 8]. When the camera is known to be stationary, background subtraction and object tracking are valuable cues (e.g., [5]). Recent developments tackle summarization for dynamic cameras that are worn or handheld [10, 8, 9] or design online algorithms to process streaming data [7].

Whereas existing methods are largely unsupervised, our idea to explicitly learn subset selection from human-given summaries is novel. Some prior work includes supervised learning *components* that are applied during selection (e.g., to generate learned region saliency metrics [8] or train classifiers for canonical viewpoints [10]), but they do not train/learn the subset selection procedure itself. Our idea is also distinct from "interactive" methods, which assume a human is in the loop to give supervision/feedback on each individual *test* video [26, 27, 12].

Our focus on the determinantal point process as the building block is largely inspired by its appealing property in modeling diversity in subset selection, as well as its success in search and ranking [17], document summarization [14], news headline displaying [28], and pose estimation [29]. Applying DPP to video summarization, however, is novel to the best of our knowledge.

Our seqDPP is closest in spirit to the recently proposed Markov DPP [28]. While both models enjoy the Markov property by defining conditional probabilities depending only on the immediate past,

Table 1: Performance of various video summarization methods on OVP. Ours and its variants perform the best.

| | Unsupervised methods | | | | Supervised subset selection | | | |
|---|---|---|---|---|---|---|---|---|
| | DT [30] | STIMO [31] | VSUMM$_1$ [24] | VSUMM$_2$ [24] | DPP + Q/D [14] | Ours (seqDPP+) | | |
| | | | | | | Q/D | LINEAR | N.NETS |
| **F** | 57.6 | 63.4 | 70.3 | 68.2 | 70.8±0.3 | 68.5±0.3 | 75.5±0.4 | **77.7**±0.4 |
| **P** | 67.7 | 60.3 | 70.6 | 73.1 | 71.5±0.4 | 66.9±0.4 | **77.5**±0.5 | 75.0±0.5 |
| **R** | 53.2 | 72.2 | 75.8 | 69.1 | 74.5±0.3 | 75.8±0.5 | 78.4±0.5 | **87.2**±0.3 |

Table 2: Performance of our method with different representation learning

| | VSUMM$_2$ [24] | | | seqDPP+LINEAR | | | seqDPP+N. NETS | | |
|---|---|---|---|---|---|---|---|---|---|
| | F | P | R | F | P | R | F | P | R |
| Youtube | 55.7 | 59.7 | 58.7 | 57.8±0.5 | 54.2±0.7 | 69.8±0.5 | **60.3**±0.5 | 59.4±0.6 | 64.9±0.5 |
| Kodak | 68.9 | 75.7 | 80.6 | 75.3±0.7 | 77.8±1.0 | 80.4±0.9 | **78.9**±0.5 | 81.9±0.8 | 81.1±0.9 |

Markov DPP's ground set is still the whole video sequence, whereas seqDPP can select diverse sets from the present time. Thus, one potential drawback of applying Markov DPP is to select video frames out of temporal order, thus failing to model the sequential nature of the data faithfully.

## 5 Experiments

We validate our approach of sequential determinantal point processes (seqDPP) for video summarization on several datasets, and obtain superior performance to competing methods.

### 5.1 Setup

**Data** We benchmark various methods on 3 video datasets: the Open Video Project (OVP), the Youtube dataset [24], and the Kodak consumer video dataset [32]. They have 50, 39[2], and 18 videos, respectively. The first two have 5 human-created summaries per video and the last has one human-created summary per video. Thus, for the first two datasets, we follow the algorithm described in section 3.1 to create an oracle summary per video. We follow the same procedure as in [24] to preprocess the video frames. We uniformly sample one frame per second and then apply two stages of pruning to remove uninformative frames. Details are in the supplementary material.

**Features** Each frame is encoded with an $\ell$2-normalized 8192-dimensional Fisher vector $\phi_i$ [33], computed from SIFT features [34]. The Fisher vector represents well the visual appearance of the video frame, and is hence used to compute the pairwise correlations of the frames in the quality-diversity decomposition (cf. eq. (4)). We derive the quality features $x_i$ by measuring the representativeness of the frame. Specifically, we place a contextual window centered around the frame of interest, and then compute its mean correlation (using the SIFT Fisher vector) to the other frames in the window. By varying the size of the windows from 5 to 15, we obtain 12-dimensional contextual features. We also add features computed from the frame saliency map [35]. To apply our method for learning representations (cf. section 3.3), however, we do not make a distinction between the two types, and instead compose a feature vector $f_i$ by concatenating $x_i$ and $\phi_i$. The dimension of our linear transformed features $W f_i$ is 10, 40 and 100 for OVP, Youtube, and Kodak, respectively. For the neural network, we use 50 hidden units and 50 output units.

**Other details** For each dataset, we randomly choose 80% of the videos for training and use the remaining 20% for testing. We run 100 rounds of experiments and report the average performance, which is evaluated by the aforementioned F-score, Precision, and Recall (cf. section 3.1). For evaluation, we follow the standard procedure: for each video, we treat each human-created summary as golden-standard and assess the quality of the summary output by our algorithm. We then average over all human annotators to obtain the evaluation metrics for that video.

### 5.2 Results

We contrast our approach to several state-of-the-art methods for video summarization—which include several leading unsupervised methods—as well as the vanilla DPP model that has been successfully used for document summarization but does not model sequential structures. We compare the methods in greater detail on the OVP dataset. Table 1 shows the results.

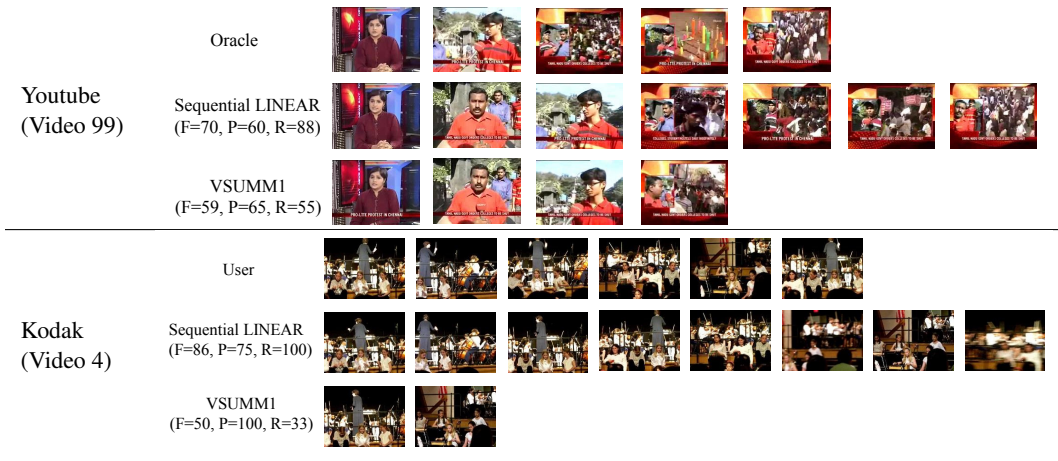

Figure 3: Exemplar video summaries by our seqDPP LINEAR vs. VSUMM summary [24].

**Unsupervised or supervised?** The four unsupervised methods are DT [30], STIMO [31], VSUMM$_1$ [24], and VSUMM$_2$ with a postprocessing step to VSUMM$_1$ to improve the precision of the results. We implement VSUMM ourselves using features described in the orignal paper and tune its parameters to have the best test performance. All 4 methods use clustering-like procedures to identify key frames as video summaries. Results of DT and STIMO are taken from their original papers. They generally underperform VSUMM.

What is interesting is that the vanilla DPP does not outperform the unsupervised methods, despite its success in other tasks. On the other end, our supervised method seqDPP, when coupled with the linear or neural network representation learning, performs significantly better than all other methods.

We believe the improvement can be attributed to two factors working in concert: (1) modeling sequential structures of the video data, and (2) more flexible and powerful representation learning. This is evidenced by the rather poor performance of seqDPP with the quality/diversity (Q/D) decomposition, where the representation of the items is severely limited such that modeling temporal structures alone is simply insufficient.

**Linear or nonlinear?** Table 2 concentrates on comparing the effectiveness of these two types of representation learning. The performances of VSUMM are provided for reference only. We see that learning representations with neural networks generally outperforms the linear representations.

**Qualitative results** We present exemplar video summaries by different methods in Fig. 3. The challenging Youtube video illustrates the advantage of sequential diverse subset selection. The visual variance in the beginning of the video is far greater (due to close-shots of people) than that at the end (zooming out). Thus the clustering-based VSUMM method is prone to select key frames from the first half of the video, collapsing the latter part. In contrast, our seqDPP copes with time-varying diversity very well. The Kodak video demonstrates again our method's ability in attaining high recall when users only make diverse selections locally but not globally. VSUMM fails to acknowledge temporally distant frames can be diverse despite their visual similarities.

## 6    Conclusion

Our novel learning model seqDPP is a successful first step towards using human-created summaries for learning to select subsets for the challenging video summarization problem. We just scratched the surface of this fruit-bearing direction. We plan to investigate how to learn more powerful representations from low-level visual cues.

**Acknowledgments** B. G., W. C. and F. S. are partially supported by DARPA D11-AP00278, NSF IIS-1065243, and ARO #W911NF-12-1-0241. K. G. is supported by ONR YIP Award N00014-12-1-0754 and gifts from Intel and Google. B. G. and W. C. also acknowledge supports from USC Viterbi Doctoral Fellowship and USC Annenberg Graduate Fellowship. We are grateful to Jiebo Luo for providing the Kodak dataset [32].

## Footnotes

[1]After all, not *all* videos on YouTube are about cats.

[2]In total there are 50 Youtube videos. We keep 39 of them after excluding the cartoon videos.

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
