[Supplementary Material · supp.pdf]

# Supplementary Material of Diverse Sequential Subset Selection for Supervised Video Summarization

**Boqing Gong**[*]
Department of Computer Science
University of Southern California
Los Angeles, CA 90089
boqinggo@usc.edu

**Wei-Lun Chao**[*]
Department of Computer Science
University of Southern California
Los Angeles, CA 90089
weilunc@usc.edu

**Kristen Grauman**
Department of Computer Science
University of Texas at Austin
Austin, TX 78701
grauman@cs.utexas.edu

**Fei Sha**
Department of Computer Science
University of Southern California
Los Angeles, CA 90089
feisha@usc.edu

We provide extra details on the following:

- Sec. A: how to use MLE to learn the model parameters of our seqDPP and the gradients used to maximize the log-likelihood.
- Sec. B: more details about the video preprocessing used in our experiments.

## A  Learning seqDPP via MLE

We learn the model parameters of our seqDPP through maximum likelihood estimation (MLE) and search for local optimums of the data log-likelihood by gradient descent. For convenience, we consider only one training video in the following derivation.

### A.1  Maximum likelihood estimation (MLE)

Recall that we have partitioned a long video sequence $\mathcal{Y}$ to $T$ disjoint short segments, $\bigcup_{t=1}^{T} \mathcal{Y}_t = \mathcal{Y}$, and defined a variable $Y_t$ at each time $t$. The corresponding oracle summary is therefore also partitioned to $T$ sets $\boldsymbol{y}_1, \cdots, \boldsymbol{y}_T$, where $\boldsymbol{y}_t \subseteq \mathcal{Y}_t$ and $\boldsymbol{y}_t$ may be an empty set. We estimate our model parameters $\Theta$ via maximizing the following log-likelihood,

$$\mathcal{L}\left(\Theta; \{(\mathcal{Y}_t, \boldsymbol{y}_t)\}_{t=1}^{T}\right) \triangleq \log P(Y_1 = \boldsymbol{y}_1, \cdots, Y_T = \boldsymbol{y}_T; \Theta) \tag{1}$$

$$= \sum_{t=1}^{T} \log P(Y_t = \boldsymbol{y}_t | Y_{t-1} = \boldsymbol{y}_{t-1}; \Theta) \tag{2}$$

where we introduce a dummy segment $\mathcal{Y}_0 = \boldsymbol{y}_0 = \emptyset$.

### A.2  Optimization

We use gradient descent to maximize the log-likelihood $\mathcal{L}\left(\Theta; \{(\mathcal{Y}_t, \boldsymbol{y}_t)\}_{t=1}^{T}\right)$ with respect to $\Theta$.

The following notation is useful to deriving the gradients $\frac{\partial \mathcal{L}}{\partial \Theta}$,

$$\mathcal{J}^{t}(\Theta; \mathcal{Y}_t, \boldsymbol{y}_t, \boldsymbol{y}_{t-1}) \triangleq \log P(Y_t = \boldsymbol{y}_t | Y_{t-1} = \boldsymbol{y}_{t-1}; \Theta) \tag{3}$$

$$= \log \det(\boldsymbol{\Omega}_{\boldsymbol{y}_{t-1} \cup \boldsymbol{y}_t}) - \log \det(\boldsymbol{\Omega}_t + \boldsymbol{I}_t), \tag{4}$$

---

[*]Equal contribution

where the DPP kernel $\mathbf{\Omega}_t = \mathbf{\Omega}_t(\Theta, \mathcal{Y}_t, \boldsymbol{y}_{t-1})$ is constructed from the data $\mathcal{Y}_t \cup \boldsymbol{y}_{t-1}$ and the model parameters $\Theta$ (cf. Section 3.2 in the main text).

Therefore, we have

$$\frac{\partial \mathcal{L}}{\partial \Theta} = \sum_{t=1}^{T} \frac{\partial \mathcal{J}^t}{\partial \Theta} = \sum_{t=1}^{T} \sum_{ij} \frac{\partial \mathcal{J}^t}{\partial \mathbf{\Omega}_{t_{ij}}} \frac{\partial \mathbf{\Omega}_{t_{ij}}}{\partial \Theta}, \tag{5}$$

*i.e.*, the gradients with respect to the model parameters are decomposed to two parts through the chain rule. One part is about the DPP kernel $J_{ij}^t \triangleq \frac{\partial \mathcal{J}^t}{\partial \mathbf{\Omega}_{t_{ij}}}$ and the other is of the DPP kernel about the model parameters $\frac{\partial \mathbf{\Omega}_{t_{ij}}}{\partial \Theta}$. The former remains the same for whatever parameterizations we use to construct the DPP kernels. It is readily computable,

$$\frac{\partial \mathcal{J}^t}{\partial \mathbf{\Omega}_t} = \frac{\partial \log \det(\mathbf{\Omega}_{\boldsymbol{y}_{t-1} \cup \boldsymbol{y}_t})}{\partial \mathbf{\Omega}_t} - \frac{\partial \log \det(\mathbf{\Omega}_t + \boldsymbol{I}_t)}{\partial \mathbf{\Omega}_t} = \mathcal{M}\left((\mathbf{\Omega}_{\boldsymbol{y}_{t-1} \cup \boldsymbol{y}_t})^{-1}\right) - (\mathbf{\Omega}_t + \boldsymbol{I}_t)^{-1} \tag{6}$$

where the operator $\mathcal{M}(\cdot)$ maps a square submatrix $\boldsymbol{A_y}$ to a matrix $\boldsymbol{B}$, such that 1) $\boldsymbol{B}$ is the same size as the orignal matrix $\boldsymbol{A}$, and 2) the sqaure submatrix $\boldsymbol{B_y} = \boldsymbol{A_y}$ and all the other entries of $\boldsymbol{B}$ are zeros.

The latter, $\frac{\partial \mathbf{\Omega}_{t_{ij}}}{\partial \Theta}$, depends on the particular forms of parameterizing the DPP kernel $\mathbf{\Omega}_t$. We provide the details with respect to the nonlinear hidden representation (cf. Section 3.3 in the main text). For convenience, we drop the subscript $t$ in what follows.

Recall that in the nonlinear hidden representation,

- $\boldsymbol{z}_i = \tanh(\boldsymbol{U}\boldsymbol{f}_i)$,
- $\mathbf{\Omega}_{ij} = \boldsymbol{z}_i^T \boldsymbol{W}^T \boldsymbol{W} \boldsymbol{z}_j$.

Immediately, we have $\frac{\partial \mathbf{\Omega}_{ij}}{\boldsymbol{W}} = \boldsymbol{W}(\boldsymbol{z}_i \boldsymbol{z}_j^T + \boldsymbol{z}_j \boldsymbol{z}_i^T)$, as well as

$$\frac{\partial \mathcal{J}}{\partial \boldsymbol{W}} = \sum_{ij} \frac{\partial \mathcal{J}}{\partial \mathbf{\Omega}_{ij}} \frac{\partial \mathbf{\Omega}_{ij}}{\boldsymbol{W}} = \sum_{ij} J_{ij} \boldsymbol{W}(\boldsymbol{z}_i \boldsymbol{z}_j^T + \boldsymbol{z}_j \boldsymbol{z}_i^T) = 2\boldsymbol{W}\boldsymbol{Z}\boldsymbol{J}\boldsymbol{Z}^T, \tag{7}$$

where $\boldsymbol{Z}$ is the column-wise concatenation of $\{\boldsymbol{z}_i\}$.

Moreover,

$$\frac{\partial \mathbf{\Omega}_{ij}}{\partial \boldsymbol{z}_i} = \boldsymbol{W}^T \boldsymbol{W} \boldsymbol{z}_j \triangleq \boldsymbol{M}\boldsymbol{z}_j, \quad \Longrightarrow \quad \frac{\partial \mathbf{\Omega}_{ij}}{\partial z_{il}} = \boldsymbol{M}^l \boldsymbol{z}_j, \tag{8}$$

where $\boldsymbol{M} = \boldsymbol{W}^T \boldsymbol{W}$ and $\boldsymbol{M}^l$ is the $l$-th row of $\boldsymbol{M}$.

Introduce $\boldsymbol{h}_i = 2\boldsymbol{f}_i$ and the inverse logit function $\sigma(x) = (1 + e^{-x})^{-1}$. Then we have

$$\boldsymbol{z}_i = \tanh(\boldsymbol{U}\boldsymbol{f}_i) = 2\sigma(\boldsymbol{U}\boldsymbol{h}_i) - 1 \tag{9}$$

$$\frac{\partial z_{il}}{\partial \boldsymbol{U}^l} = \frac{\partial [2\sigma(\boldsymbol{U}^l \boldsymbol{h}_i) - 1]}{\partial \boldsymbol{U}^l} = 2\sigma(\boldsymbol{U}^l \boldsymbol{h}_i)[1 - \sigma(\boldsymbol{U}^l \boldsymbol{h}_i)]\boldsymbol{h}_i^T \triangleq s_{il}\boldsymbol{h}_i^T, \text{ and} \tag{10}$$

$$\frac{\partial \mathbf{\Omega}_{ij}}{\boldsymbol{U}^l} = \frac{\partial \mathbf{\Omega}_{ij}}{\partial z_{il}} \frac{\partial z_{il}}{\partial \boldsymbol{U}^l} + \frac{\partial \mathbf{\Omega}_{ij}}{\partial z_{jl}} \frac{\partial z_{jl}}{\partial \boldsymbol{U}^l} = \boldsymbol{M}^l \boldsymbol{z}_j s_{il} \boldsymbol{h}_i^T + \boldsymbol{M}^l \boldsymbol{z}_i s_{jl} \boldsymbol{h}_j^T \tag{11}$$

Overall, we have the gradients with respect to $\boldsymbol{U}$ in the following,

$$\frac{\partial \mathcal{J}}{\partial \boldsymbol{U}^l} = \sum_{ij} \frac{\partial \mathcal{J}}{\partial \mathbf{\Omega}_{ij}} \frac{\partial \mathbf{\Omega}_{ij}}{\partial \boldsymbol{U}^l} = \sum_{ij} J_{ij}(\boldsymbol{M}^l \boldsymbol{z}_j s_{il} \boldsymbol{h}_i^T + \boldsymbol{M}^l \boldsymbol{z}_i s_{jl} \boldsymbol{h}_j^T) = 2\boldsymbol{M}^l \boldsymbol{Z}\boldsymbol{J}\text{DIAG}(s_{1l}, s_{2l}, \cdots)\boldsymbol{H}^T, \tag{12}$$

where $\boldsymbol{H}$ is the column-wise concatenation of $\{\boldsymbol{h}_i\}$.

## B    Preprocessing of the video frames

We preprocess the videos from OVP and Youtube datasets as follows. First, we uniformly sample one frame per second. We then remove the transitional frames which are close to the shot boundaries, since these are likely dissolving, wiping, or fading frames with confusing content or little information. Finally, we remove near-monotone frames by calculating the entropy of the color histogram of each frame. The resulting ground set is of average size 84 and 128 for the OVP and Youtube videos, respectively. The Kodak dataset consists of consumer videos that are mainly with a single shot. Therefore, after uniform sampling (two frames per second considering the Kodak videos are often short, around 1 minute), no further pruning is applied. On average 50 frames are kept for the Kodak videos after preprocessing. For seqDPP, we further partition the preprocessed video (ground sets) into consecutive segments, each of which has $n = 10$ frames (seqDPP is robust when $n$ varies from 10 to 20). We select the smallest $n$ so as to have efficient inference for each segment.