[Reviews · NeurIPS 2014]

Submitted by Assigned_Reviewer_38

The paper proposes an approach to find a diverse subset of frames to summarize a video clip, by using an extension of the determinantal point process framework. The paper is well written and easy to read.

The authors propose a very simple extension of the DPP work. It would have been more interesting if there was some discussion on how the size of the subset can also be controlled in the formulation. As can be seen from the results in Fig3, the proposed method returns more frames than the oracle. So it would be interesting to see how the method can be adapted for subsets of a fixed size.

Other points:
1) line 247, how is "partitioning into T disjoint segments" accomplished? Provide details.
2) Line 256, it is not clear why the diagonal matrix I_t has zeros corresponding to the elements of y_t and 1s for elements of Y_t. need to add explanation.
3) why are there only 2 frames for vsumm? cant you apply vsumm to the separate video segments obtained in Line 256 separately?
A simple baseline i would like to see results for is to choose one frame randomly from each segment
Summary: The paper is well written and clear, and proposes a simple extension of the DPP approach for video summarization by selecting diverse subsets.

Submitted by Assigned_Reviewer_39

The paper addresses the problem of video content summarization by 1) fitting a system to meet the criteria of human operators via a supervised fashion; and 2) augmenting the deterministic point process with sequential modeling.
The paper contributes at both conceptual and technical levels, and justifies these novelties with superior results on benchmark datasets over state-of-the-art unsupervised methods.
It is clearly shown that 1) optimizing the video summarization via the data-driven manner (i.e., with supervised learning) leads to a better scheme than using hand crafted criteria; and 2) temporal structure of video plays a crucial role in video summarization.

Clarity:
The paper is well written and easy to understand.
Summary: The paper addresses the problem of video summarization with insightful motivation and solid technical solution. The empirical result justifies the effectiveness of the proposed method on benchmarks.

Submitted by Assigned_Reviewer_41

This paper presents a framework for supervised video summarization. The proposed method includes a new Sequential Determinantal Point Process (seqDPP) model, which learns to select the best subsets of the raw input data as components of the output summarization. Unlike the most common approaches, that focus on unsupervised summarization by selecting most important/representative frames and features, this paper attempts to directly learn from human-created summaries.

The technical formulation of the method seems sound and correct. The new model seqDPP extends DPP by incorporating sequential dependencies. The model, learning and inference formulations are updated to account for the extension.

The performance evaluation is adequate, including quantitative results (Precision, Recall and F-Score) from 3 datasets (youtube, OVP, kodak consumer). The results are good and mostly outperform the related work.

The readability of the document is good. However the order of the sections could be improved. For instance, the Related Work could be presented earlier to help understand the context of this paper.

In terms of originality, the seqDPP model is an extension of DPP which is used to summarize text data. The extension is sound to tackle the problem of video summarization. Also, the authors use a novel way to derive ground truth, by combining multiple human-created summaries into a unique ‘oracle’ summary, which is used for learning and evaluation.
Summary: The authors present a novel model, with an appropriate evaluation getting good results which mostly outperforms related work.
Author Feedback
Author rebuttal: We thank all reviewers for their comments. We are pleased to see that both R39 and R41 positively appreciate the novelty of the proposed approach (seqDPP), as well as the superior experimental results.

The developed technique in this paper is an important first step into a previously uncharted but fruitful direction of supervised learning for video summarization. Besides the conceptual novelty, our technical advancements include adapting DPPs for modeling videos (and potentially other types of sequential data).

We focus on the technical questions raised by R38, recognizing that the other two reviewers are in general satisfied with the technical and conceptual expositions in this paper.

Q: Controlling the size of the subset to be selected.

A: The method of seqDPP is flexible such that the size of the subset can be controlled within each segment. For instance, analogous to k-DPP[17], we can extend straightforwardly to cap the size to at most k. This will restrict the total number of frames extracted to be no more than k*M, where M is the number of segments.

To constrain the total number of frames but avoid imposing constraints individually on each segment would require more detailed modeling and analysis. We leave that for future work. (A possible lead is to use dynamic programming to maximize the total "score" contributed additively by each segment, resulting in different numbers of frames.)

Q: How is "partitioning into T disjoint segments" accomplished?

A: We partition the (post-processed) video into consecutive segments, each of which has n=10 frames. (The algorithm is robust when n varies from 10 to 20). We select the smallest n so as to have efficient inference for each segment.

Q: Line 256, it is not clear why the diagonal matrix I_t has zeros corresponding to the elements of y_t and 1s for elements of Y_t. need to add explanation.

A: It follows from the algebraic properties of DPP. Namely, it ensures that eq. (7) is a valid probability distribution. We will provide a more detailed derivation in the paper.

Q: Why are there only 2 frames for vsumm (in Fig. 3)? Applying vsumm to the separate video segments?

A: Fig. 3 clearly illustrates the weakness of VSUMM: it ignores the temporal nature of the frames and clusters them as if they were independent. Thus, it fails on videos that are not diverse globally but vary significantly locally.

Applying VSUMM to each segment is difficult and likely not useful in practice. It would require a significantly increased number of heuristics for tuning various (hyper)parameters: to control the size of the subset for each segment, to adapt thresholding across segments, etc. Moreover, VSUMM selects at least one (and usually more) summary frame per segment. Thus it will always yield more frames than the oracle.

Q: A simple baseline I would like to see results for is to choose one frame randomly from each segment

A: We ran the suggested baseline for 100 random rounds of experiments following the setup in Section 5. The F-scores of this baseline on OVP, Youtube, and Kodak are 57.6+-0.3, 49.1+-0.4, and 68.6+-0.7, respectively. They are much worse than the F-scores of seqDPP (77.7+-0.4 , 60.3+-0.5, and 78.9+-0.5).